# Spatial Frequency Response of Epoxy-Based Volume Holographic Recording Material

**DOI:** 10.3390/molecules24061018

**Published:** 2019-03-14

**Authors:** Tina Sabel

**Affiliations:** Department of Chemistry, Technische Universität Berlin, Strasse des 17. Juni 135, 10623 Berlin, Germany; tina@physik.tu-berlin.de

**Keywords:** photosensitive materials, volume holographic phase gratings, diffractive optical elements, material response, secondary gratings, transmission gratings, reflection gratings

## Abstract

Holographic volume phase gratings are recorded in an epoxy-based, free-surface, volume holographic recording material. Light-induced gratings are formed by photo-triggered mass migration caused by component diffusion. The material resolution enables a wide range of pattern spacings, to record both transmission and reflection holograms with many different spatial frequencies. An optimum spatial frequency response is found between the low spatial frequency roll-off and the high spatial frequency cut-off. The influence of the energy density of exposure on the spatial frequency response is investigated. Secondary volume holographic gratings (parasitic gratings) are observed in the high frequency range. The possibility of distinguishing the regular grating from the secondary grating is discussed in the form of probe wavelength detuning.

## 1. Introduction

Photosensitive polymers represent a particularly interesting group among stimuli-responsive polymeric materials, standing out due to their ability to be applied in a non-invasive and easily controlled manner [1]. Light as a stimuli entails optical structuring by the application of volume holography as a single-step method for manufacturing 3D diffractive micro- and nanostructures [2]. Some of the many possible uses for volume holographic materials are as diffractive optical elements for applications related to security purposes and holographic-based memories, self-written waveguides, wavelength-selective devices, solar energy concentrators, and multifunctional materials [3,4,5,6].

Volume holographic grating formation in photosensitive polymers can be attributed to an interplay of polymerization and diffusion, induced by spatially modulated exposure. A light pattern is projected into the photosensitive medium, inducing local polymerization proportional to the light intensity. Thereupon, a chemical gradient is induced, resulting in monomer diffusion and subsequent polymerization. As a consequence, the hologram is formed as a periodic modulation of optical properties—usually the refractive index—according to the recording light pattern [4,7]. For an illustration of the grating formation mechanism see Section 3. Materials and Methods. The final grating is characterized by the grating period, the refractive index contrast, and the shape of the modulation profile [8].

The spatial frequency of a grating results from the grating period, which is determined by the interference pattern, defined by the exposure geometry:(1)$$\Lambda =\frac{\lambda }{2n\sin\theta /2},$$
where *n* is the average refractive index of the medium, *λ* is the wavelength of exposure, and *θ* is the angle enclosed by the two recording beams in the air. Many applications require large refractive index modulations at spatial frequencies in the order of 1000 lines/mm [9,10]. Realizing a material system that is capable of recording a wide range of pattern spacings from low to high spatial frequency poses a challenge with regard to the material composition [11,12].

Recently, a novel epoxy-based material for volume holography was introduced, standing out due to its high diffraction efficiency and dimensional stability, low shrinkage, and great resistance to environmental conditions, having the special feature of a free surface [13].

Developing a novel material entails the optimization of the material response, which refers to the ability of a photosensitive material to transfer the interference pattern of exposure into a permanent modulation of optical properties during the recording process. Analysis of the material response comprises different investigations, such as of the energetic sensitivity and angular response [4]. Those characteristics are determined by the influence of the energy density of exposure on the final contrast of the holographic grating. Results of the material response of the epoxy-based material in terms of energetic sensitivity and angular response have already been reported [13]. The present paper concentrates on the spatial frequency response, i.e., the impact of the grating period on the grating formation, as an important material characteristic and a key criterion for the scope of possible applications for the material [14]. The spatial frequency is one of many internal and external factors, such as the material’s viscosity and the exposure duration, that show an influence on the different aspects of the material performance, such as material response and optical shrinkage [15,16].

The course of the spatial frequency response function reveals important material characteristics. First, a high frequency cut-off is expected, i.e., a decrease in the diffraction efficiency for high spatial frequencies, as a result of real exposure conditions and non-local spatial response effects. Possible sources of contrast loss are the mechanical instabilities of the recording setup. In particular, long exposure durations must result in a cumulative smearing of the grating profile. On the other hand, it is reasonable to expect a fundamental physical limit for the minimum recordable grating period, depending on the recording mechanism [17]. As the spatial frequency response is a characteristic of the specific material, the response at a high spatial frequency can be improved by means of an adjustment of the composition [18]. A characteristic feature is the maximum spatial frequency response, i.e., the highest frequency inducing a permanent refractive index modulation. This marks the maximum achievable density of interference fringes that can be recorded in the photosensitive material. In terms of holographic data storage, this corresponds to the achievable storage density. A high material resolution is particularly required for reflection-type holograms. It is also accompanied by the highest possible variability in angular selectivity, with corresponding effects on the diffractive properties. A resolution of 5000–10,000 lines/mm is aspired to for most applications. In view of the high resolution nanostructures, achievable with material based on SU-8 photoresist [19], the material resolution of the epoxy-based material is expected to be suitable for volume holographic recording. Beyond the limit, which arises from specific material properties, the maximum spatial frequency is a priori restricted by means of the fabrication technique. The wavelength of exposure determines the smallest feature size to *λ*/2*n*.

Furthermore, the diffusion model predicts a low spatial frequency roll-off as a consequence of the ratio of the rate of diffusion to the rate of polymerization. The decreasing material response at a low spatial frequency arises because the *R* parameter is inversely proportional to the square of the grating period Λ [8,13,20]:(2)$$R=\frac{4\pi^{2}D}{\Lambda^{2}F_{0}},$$
with the diffusion coefficient *D* and half the maximum polymerization rate *F*_0_.

Thus, the low spatial frequency roll-off can be observed as a consequence of insufficient monomer diffusion across the larger grating periods [21]. With respect to the diffusion model, both phenomena in the spatial frequency response can be deduced, introducing a Gaussian non-local material response function into the diffusion equation [22].

Finally, it is expected that an optimum spatial frequency response will be found between the low spatial frequency roll-off and the high spatial frequency cut-off [21]. Here, there is a good linearity of the transfer to diffraction orders, i.e., a constant index contrast for the respective grating constant results in the highest diffraction efficiency. According to appropriate models for the free-radical polymerization, the optimal spatial frequency response is expected to be between 500 and 1000 lines/mm [23]. In the case of nanoparticle-dispersed acrylate photopolymers, the optimum (Δ*n* = 5 ⋯ 10^−3^) is found at Λ = 1.5 µm, i.e., 700 lines/mm [24]. For the Du Pont polymer, the optimum is located at around 2000 lines/mm [25]. For acrylamide-based photopolymers, lower spatial frequencies (a few hundred lines per millimeter) are preferable for applications as multiplexed or stacked gratings for solar collectors [26], while the optimum is located between 650 and 1000 lines/mm [27]. Epoxy-based materials differ with respect to the mechanism of polymerization [28,29,30]. Holograms are formed in an organic cationic ring-opening polymerization. This is also expected to have certain effects on the spatial frequency response.

## 2. Results and Discussion

### 2.1. Spectral Width of the Material Response

The material under investigation was designed to record a wide range of pattern spacings, enabling both transmission- and reflection-type holograms. Figure 1 shows light-microscopic images of a transmission grating and a reflection grating.

The two structures differ significantly with respect to their spatial frequency. The transmission grating in Figure 1a has a grating period of Λ*_T_* = 2.5 µm. This corresponds to a spatial frequency of 400 lines/mm. The reflection grating is displayed in a sectional view. Imaging in sectional view was implemented for the analysis of reflection-type gratings with periods far below the resolution limit of optical microscopy [31]. Hereby, a tilted reflection-type grating appears in the image plane as a sectional grating with a much larger periodicity. Figure 1b shows such an image in sectional view. The sectional grating period, derived from Figure 1, is Λ*_S_* = 30 µm. This corresponds to the real grating period of Λ*_R_* = 130 nm (the slant angle is 0.25 degrees) and to a spatial frequency of about 8000 lines/mm. This represents the highest possible frequency within the limits of the fabrication technique.

### 2.2. The Optimum Spatial Frequency Response

Figure 2 shows the results of the spatial frequency response for transmission-type gratings. The applied variation of the grating period from 600 nm up to 5 µm corresponds to a range that is readily accessible within transmission exposure geometry.

The optimum material response can be derived from Figure 2. Grating periods around Λ = 3 µm, i.e., approximately 350 lines/mm, resulted in the highest diffraction efficiency. Thus, the optimum spatial frequency response of the epoxy-based material is comparable to that of acrylamide-based photopolymers [26]. With respect to the influence of the exposure dose, Figure 2 reveals that the optimum spatial frequency response depends on the energy density of the exposure—a decrease in the exposure dose results in a shift of the optimum spatial frequency response towards larger grating periods. In the case of the low-energy response curve (0.6 J/cm^2^, blue curve in Figure 2), the optimum is not even achieved within the illustrated range and thus is expected to be above 5 µm. In the high spatial frequency range, increasing the energy density of the exposure results in a higher diffraction efficiency, while for the low spatial frequency range it can be stated that a lower energy density of exposure results in a higher optimal material response. Since the low dose is achieved by means of a short exposure, this is in compliance with preliminary results on the performance of this material, indicating that a short exposure duration is preferable for the material response [13].

Independent of the recording energy density, the high spatial frequency cut-off is much more pronounced than the low frequency roll-off. This implies linear refractive index profiles in the case of big grating periods. Monomer diffusion appears to be sufficiently fast. In the case of small grating periods, the decrease in the material response may be caused by a smearing of the grating profile, as a result of insufficient vibration isolation of the recording setup. However, as discussed above, the high spatial frequency cut-off may also result from the fundamental limit of the material system [17].

### 2.3. The Maximum Spatial Frequency Response

The decrease in the material response for high spatial frequencies involves some challenges with respect to the analysis of high-resolution structures. As a result of the significantly better material response in the case of low frequencies, the arising secondary transmission gratings are predominant. Due to the indistinguishability of the Bragg angles, a separation of the reflection grating from the secondary transmission grating cannot be achieved if probing with the wavelength of exposure, i.e., for *λ_p_* = *λ_exp_*.

One possibility for avoiding secondary grating formation is to apply a single-beam exposure [31]. This technique was applied in the case of the reflection grating shown in Figure 1b. In this case, the grating period is determined by the incidence angle of the recording beam only. According to the suboptimal visibility of the interference pattern, a false-color display was used to improve the visibility of the grating in the microscopic image. As a drawback of this method, the low visibility of the interference pattern results in suboptimal contrast. This is contrary to the aim of achieving optimal material response with the highest possible diffraction efficiency.

However, a single beam exposure with perpendicular incidence (*θ* = 0°) enables the verification of the resolution ability of the system. A single beam interfering with its own reflection, induces the smallest feature size grating without secondary gratings, in absolute contrast to the two-beam exposure in 180° geometry. Investigations of the single beam exposures reveal that the material resolution of the epoxy-based system enables three-dimensional periodic structuring with at least 8000 lines/mm. This corresponds to the maximum frequency possible within the applied fabrication technique.

### 2.4. Separation of High Spatial Frequency Gratings

Beyond the demonstration of feasibility for the resolution of the maximum possible frequency, it is most important to obtain information on the quality of the high-resolution structure. Here, we encounter the problem of secondary grating formation, i.e., unintentionally recorded structures arising from interference from the incident reference beam and the reflected signal beam [31]. The possibility of separating the high-resolution reflection grating from the secondary transmission grating arises by the detuning of the probe wavelength.

Exploiting the wavelength dependency of the Bragg angles enables the two gratings to be distinguished. The reflection grating and the corresponding secondary transmission grating differ with respect to the grating period as well as the grating slant angle. As a consequence, the Bragg angles are shifted to a different extent and in opposite directions under the variation of the probe wavelength. A small detuning of the probe wavelength of only a few nanometers is sufficient to separate the reflection grating from the secondary transmission grating. Figure 3 shows the corresponding results.

The calculation is based on Equation (1). The corresponding parameters are *λ_exp_* = 405.7 nm for the wavelength of exposure and *θ* = 20° with regard to the exposure geometry, resulting in a transmission grating period of Λ*_T_* = 1.18 µm and a reflection grating period of Λ*_R_* = 126.8 nm. The indistinguishability of the Bragg angles in the event of probing with the wavelength of exposure (*λ_p_* = *λ_exp_*) is also apparent from Figure 3.

The approach of probe wavelength detuning opens the possibility of characterizing the two gratings separately. Thus, it is possible to obtain information on the reflection grating in the high frequency range without disturbance effects from the secondary (parasitic) transmission grating in the low spatial frequency range.

## 3. Materials and Methods

### 3.1. Sample Preparation

Free-surface, ultraviolet curable epoxy-based samples were prepared by micro resist technology GmbH. Both host and guest molecules featured epoxy functional groups with the corresponding mechanism of cationic ring-opening polymerization. The refractive indices of the host and guest components at the sodium-d line were *n_host_* > 1.58 and *n_guest_* ≈ 1.46, respectively. A sensitized photoacid generator (PAG) was used to induce cross-linking by cationic polymerization at 405 nm [5]. For a schematic illustration of the host–guest material composition, see Figure 4a.

Spin coating of the material in a solution of glass substrates with a rotation speed of 800 min^−1^ resulted in a layer thickness of 200 µm. A subsequent pre-exposure bake was carried out on a hotplate (80 °C) for 30 min. This step drove out the remaining solvent in order to obtain a tack-free film.

More details on the host–guest system, in terms of composition as well as performance—such as energetic sensitivity and angular selectivity—are given in [13].

### 3.2. Holographic Exposure

All investigations were based on one-dimensional, plane-wave, transmission- as well as reflection-type volume holographic gratings. Symmetric recording geometry resulted in unslanted gratings, with a periodicity of Λ*_T_* = 2.5 µm for transmission gratings and Λ*_R_* = 130 nm for reflection gratings.

Holographic exposure was performed by two freely propagating, s-polarized recording beams with a wavelength of *λ_exp_* = 405 nm and a beam diameter of 2 mm. A schematic illustration of the holographic exposure setup for transmission geometry is shown in Figure 4b. To record reflection-type holograms, the sample was rotated by 90 degrees. After the completion of holographic grating formation, samples were fixed by a UV flood cure with a dose of 350 mJ/cm^2^. During this curing step the remaining photoinitiator was used up, resulting in a sample which was no longer light-sensitive. No postbake, hardbake, or any additional developing was applied.

### 3.3. Hologram Characterization

Analysis of the final holograms was accomplished in a rotation-scan setup with a collimated probe beam. The rotation-scan setup is shown in Figure 4c. The transmitted signal of a HeNe laser (*λ_p_* = 543 nm) was detected while the hologram under test was rotated. The diffraction efficiency was calculated from the angular resolved transmission. The grating constant Λ was derived using Equation (1). The probe beam featured a diameter of 0.2 mm. Probing only a tenth of the exposed area was primarily for the purpose of measuring precision [32].

## 4. Conclusions

Focused on a promising new organic photosensitive material, the impact of the spatial frequency of exposure on the material response has been analyzed. In addition to the advantages of the material in terms of high diffraction efficiency and dimensional stability, low shrinkage, and great resistance to environmental conditions, it also offers a great degree of flexibility for working at lower and higher spatial frequencies, both in transmission and reflection geometries. Accordingly, the epoxy-based host–guest material system is suitable for a wide range of applications, from diffractive optical elements in the very low spatial frequency range to reflection gratings in the high spatial frequency range [10]. With regard to the maximum spatial frequency response, it has been shown that the material resolution of the epoxy-based system enables three-dimensional periodic structuring with at least 8000 lines/mm. This makes it suitable for volume holographic recording. The optimum material response has proven to depend on the energy density of exposure. For an exposure dose of 1 J/cm^2^, the optimum can be found at around 350 lines/mm. Secondary transmission gratings, which accompany reflection gratings in the high frequency range, have been separated from the regular grating by a small detuning of the probe wavelength. As an outlook, investigations on the impact of the spatial frequency on dimensional changes, such as shrinkage in the photopolymer layer, could complement the results presented here [16], as well as the exploration of further possibilities for improving the response of the material at a high spatial frequency with regard to the composition of the material [18].

## Figures and Tables

**Figure 1 molecules-24-01018-f001:**
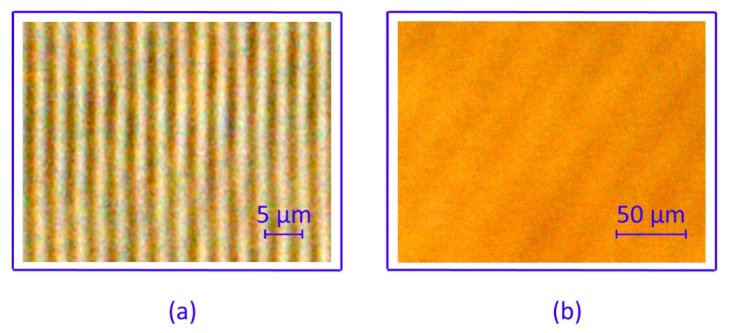
Light-microscopic image (false color display) of (**a**) transmission grating and (**b**) reflection grating in cross-sectional view.

**Figure 2 molecules-24-01018-f002:**
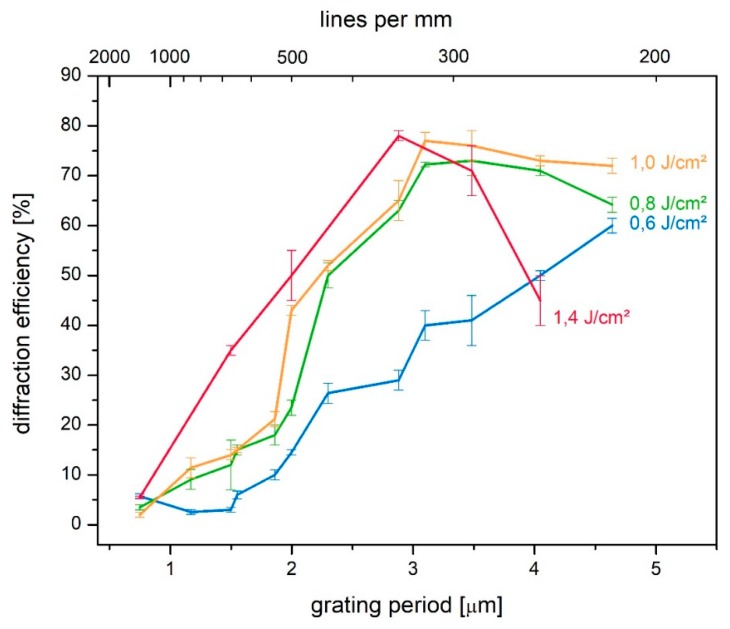
Spatial frequency response for transmission-type gratings under variation of the exposure dose.

**Figure 3 molecules-24-01018-f003:**
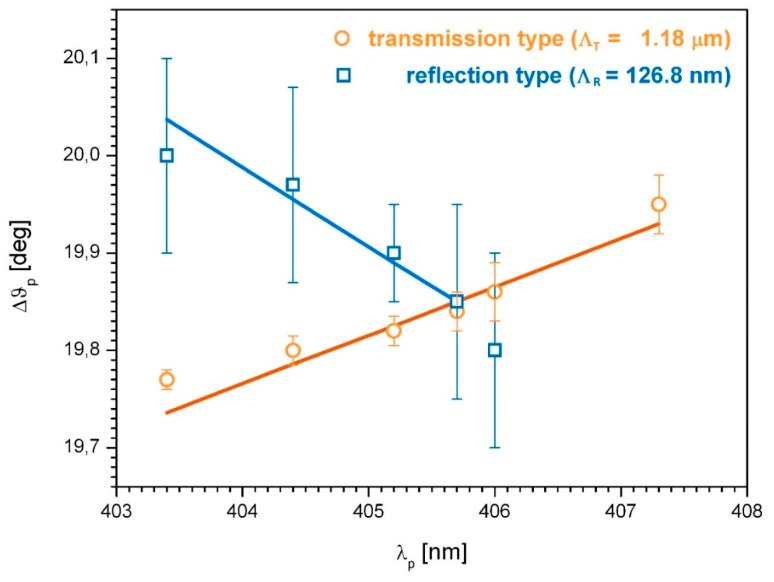
Bragg angle shift with probe wavelength detuning for reflection grating (blue squares) and transmission grating (orange circles), in comparison with calculation (lines) based on Λ*_T_* = 1.18 µm and Λ*_R_* = 126.8 nm.

**Figure 4 molecules-24-01018-f004:**
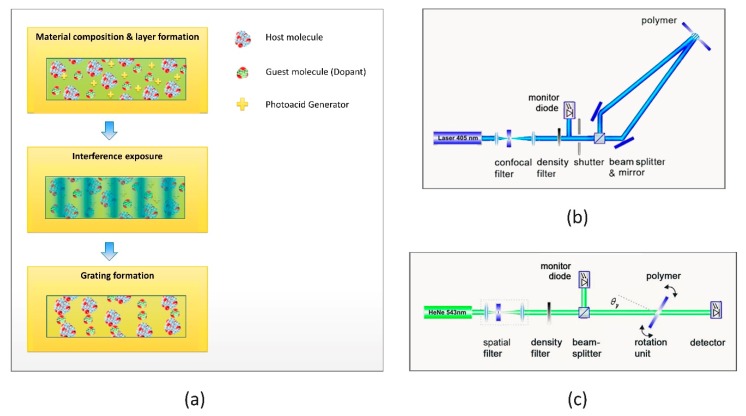
Overview on Materials and Methods: (**a**) Schematic illustration of composition, exposure, and grating formation for the host–guest material. (**b**) Holographic exposure setup with geometry for transmission gratings. (**c**) Rotation-scan setup.

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
