# Peer review of "Spatial Frequency Response of Epoxy-Based Volume Holographic Recording Material"

_molecules, 2019, doi:10.3390/molecules24061018_

Round 1
Reviewer 1 Report
1. A description of the components of the epoxy photopolymer must be included. Chemical description of the oligomers, PAG, solvent and other components used. Also the concentration used for each component. In this way other researchers could repeat the experiments and advance in the development of the material.
If there is a patent, the reference must be included. If the material is confidential know-how of the company that has provided the samples, examples published in the bibliography for similar components must be included and a name given to the material that allows its future traceability.
Author must understand that if the material is not patented or commercialized its composition (if unknown) can undergo great changes and the results of the research done will not have great value for the community because the researches could not obtain conclusions to improve the epoxy photopolymer.
2. Introduction, line 88. Results obtained from different materials which undergo free-radical polymerization are compared: acrylate photopolymers, Du Pont polymer and acrylamide based photopolymers. Why is it not compared with epoxy type photopolymers? Chemistry of acrylate and acrylamide are different from epoxy.
3. For acrylamide based photopolymers, it is said that the optimal spatial frequency is about a few hundred lines per millimeter. This could be preferable for some applications as also is remarked in the text. After results shown in Fig. 2, for the material of this work, 350 lines per mm result in the highest diffraction efficiency. "Thus, the optimum spatial frequency response of the epoxy-based material is comparable to that of acrylamide based photopolymers".
This is not exactly like this. The advantages of AA-based photopolymers are their high DE and their good energy sensitivity due to the different chemistry of these materials with respect to epoxy. The acrylamide-based photopolymers achieve a DEmax at a spatial frequency of 1000 lines / mm but also at lower and higher frequencies in transmission.
Ortuño, M., Gallego, S., García, C. et al. Appl. Phys. B (2003) 76: 851. https://doi.org/10.1007/s00340-003-1229-0
S. Gallego, M. Ortuño, C. García, C. Neipp, A. Beléndez & I. Pascual * (2005) High-efficiency volume holograms recording on acrylamide and N,N′methylene-bis-acrylamide photopolymer with pulsed laser, Journal of Modern Optics, 52:11, 1575-1584, DOI: 10.1080/09500340500058157
Holographic Characteristics of an Acrylamide/Bisacrylamide Photopolymer in 40–1000 µm Thick Layers M Ortuño, S Gallego, C García, I Pascual, C Neipp and A Beléndez
Physica Scripta, Volume 2005, T118
You can also obtain a high diffraction efficency in reflection with acrylamide based polymers:
Analysis of holographic reflection gratings recorded in polyvinyl alcohol/acrylamide photopolymer in Applied Optics 52(8):1581-90 · March 2013
For certain applications a smaller spatial frequency may be necessary but that does not mean that it is the limit of these materials as the text seems to indicate. This must be corrected.
4. The conclusion also must be improved: "...For exposure dose of 226 1 J/cm² the optimum can be found around 350 lines per mm and is therefore comparable to that of 227 acrylamide-based photopolymers."
Acrylamide photopolymers have better energetic sensitivity than epoxy because the different chemistry.
5. Author should emphasize the advantages of epoxy material: DEmax close to 80% around 350 lines / mm. Flexibility to work at lower and higher spatial frequencies, both in transmission and in reflection. Taking into account also that the material has great dimensional stability, very low shrinkage and great resistance to environmental conditions, unlike acrylamide based photopolymers.
6. Author has added 9 own works in the references. References from other authors regarding epoxy based materials are missing, so that the results obtained with the same type of material can not be compared.
7. "In case of small grating periods the decrease of the material response may be caused by a smearing of the grating profile as a result of insufficient vibration isolation of the recording setup"
Transmission: The result could be related to the chemistry of the epoxy instead of problems with vibration. For example, I have recorded holograms with high diffraction efficiency at 1000 lines/mm without air table.
Reflection: It could be a problem with the vibration as the author says. Other possibility is related with the chemistry of the material. For example with 2656 lines/mm, acrylamide based polymers have the same problem of decrease of the material response but it is solved using a chain transfer agent:
E. Fernandez, R. Fuentes, M. Ortuño, A. Beléndez and I. Pascual, "Diffraction efficiency improvement in high spatial frequency holographic gratings stored in PVA/AA photopolymers: several ACPA concentrations", Journal of Optics vol. 17, nº 1, art. 015401 (2015).
Author Response
Dear Reviewer,
Thank you very much for the valuable comments on the manuscript “Spatial frequency response of epoxy-based volume holographic recording material”, which are very helpful to improve the manuscript. I have made changes on the manuscript in order to address all the comments. Please find attached the point-by-point replies to the comments and detailed changes made in the revised manuscript.

Reviewer 2 Report
Report on the paper by Sabel
The manuscript described a technical note on holographic recording. It is not suitable for a full paper but for a technical note.
1. With regard to Fig. 2, it is required to discuss in relation to the wavelength of the light.
Author Response
Dear Reviewer,
Thank you very much for the comments on the manuscript “Spatial frequency response of epoxy-based volume holographic recording material”. I have made some changes on the manuscript in order to address all the comments. Please find below point-by-point replies to the comments and detailed changes made in the revised manuscript.
Response to comments:
The present article comprises a detailed description of the study on the spatial frequency response of the epoxy material with a clear research question and presents novel and interesting results on the material response. Thus, it contributes to new scientific knowledge and is therefore clearly a full paper and not a technical note.
Figure 2 presents results on the spatial frequency response for transmission type gratings. The gratings are recorded with wavelength 405 nm, and scanned with 543 nm, as described in the manuscript. I do not see another influence of the light wavelength.
Hopefully all your concerns have been addressed. If you still have any question, please let me know. Criticisms and suggestions are always welcome and appreciated.
Reviewer 3 Report
In this paper holographic volume gratings (in both transmission and reflection modes) are recorded in epoxy-based material and optimum spatial frequency in the view point of diffraction efficiency is studied. Although the paper is written in very comprehensive style, my overall expression is that some important details and discussions are missing. For example, in the study concerning reflection holograms no results for diffraction efficiencies or different recording angles are presented. Also the author mentions recording reflection holograms but figure 4b presents the holographic set-up for recording transmission holograms only. Further, it is not clear why grating with period of 130 nm appears as a low contrast grating with period of 30 microns in figure 1b. However, my main concerns are about the lack of optimization of the recording material. I wonder is it possible the results to be improved if some optimization of the chemical composition of the recording medium is performed.
Finally, I would like to stress author’s attention on few small things. In introduction part R-parameters is mentioned but it is not clear how is it define. Could the authors explain in more details? Also my recommendation is firstly to describe the material formulation, preparation of samples and holographic set-up and after then to present the results and discussions.
Author Response
Dear Reviewer,
thank you very much for the valuable comments on the manuscript “Spatial frequency response of epoxy-based volume holographic recording material”, which are very helpful to improve the manuscript. I have made some changes on the manuscript in order to address all the comments. Please find below point-by-point replies to the comments and detailed changes made in the revised manuscript.
Response to comments:
I belief a presentation of results for diffraction efficiencies or different recording angles for reflection holograms would go beyond the scope of the present paper.
For the holographic setup it was added to the figure caption that it displays “Holographic exposure setup with geometry for transmission gratings.” (line 231). Furthermore, it was added to line 216: that “Figure 4b [is] for transmission geometry. To record reflection type holograms, the sample would be rotated by 90 degrees.”.
The reason why the grating with period of 130 nm appears as a low contrast grating with period of 30 microns in figure 1b is given in the second paragraph on page 3: “The reflection grating is displayed in sectional view. Imaging in sectional view was implemented for analysis of reflection type gratings with periods far below the resolution limit of optical microscopy [31]. Hereby a tilted reflection-type grating appears in the image plane as a sectional grating with much larger periodicity.” Further details on sectional gratings would go beyond the scope of the present manuscript and can be found in reference 31:
31: Sabel, T.; Zschocher, M. Imaging of Volume Phase Gratings in a Photosensitive Polymer, Recorded in Transmission and Reflection Geometry. Appl. Sci 2014, 4, 19–27.
An optimization of the chemical composition of the recording medium has been performed and is reported in reference 13.
A definition of the R-parameter has been added to the introduction (line 84).
Regarding the recommendation to put the Materials and Methods Section prior to the Results and Discussion: I would also have preferred this. However, it is not possible due to the journals style standards.
Hopefully all your concerns have been addressed. If you still have any question, please let me know. Criticisms and suggestions are always welcome and appreciated.
Round 2
Reviewer 3 Report
I would like to thank to the author for the detailed answers. All my comment and amendments are taken into consideration in the revised manuscript. My recommendation is the paper to be published in the present form.